# Integrated Transcriptome and Metabolome Analyses Uncover Cholesterol-Responsive Gene Networks

**DOI:** 10.3390/ijms26157108

**Published:** 2025-07-23

**Authors:** Ruihao Zhang, Qi Sun, Lixia Huang, Jian Li

**Affiliations:** Department of Biochemistry and Molecular Biology, School of Basic Medical Sciences, Cheeloo College of Medicine, Shandong University, Jinan 250012, China; 201920593@mail.sdu.edu.cn (R.Z.); 202020695@mail.sdu.edu.cn (Q.S.); 201915061@mail.sdu.edu.cn (L.H.)

**Keywords:** transcriptomics, metabolomics, O-linked glycosylation, cholesterol

## Abstract

Cholesterol stress profoundly modulates cellular processes, but its underlying mechanisms remain incompletely understood. To investigate cholesterol-responsive networks, we performed integrated transcriptome (RNA-seq) and metabolome (LC-MS) analyses on HeLa cells treated with cholesterol for 6 and 24 h. Through transcriptomic analysis of cholesterol-stressed HeLa cells, we identified stage-specific responses characterized by early-phase stress responses and late-phase immune-metabolic coordination. This revealed 1340 upregulated and 976 downregulated genes after a 6 h cholesterol treatment, including induction and suppression of genes involved in cholesterol efflux and sterol biosynthesis, respectively, transitioning to Nuclear Factor kappa-B (NF-κB) activation and Peroxisome Proliferator-Activated Receptor (PPAR) pathway modulation by 24 h. Co-expression network analysis prioritized functional modules intersecting with differentially expressed genes. We also performed untargeted metabolomics using cells treated with cholesterol for 6 h, which demonstrated extensive remodeling of lipid species. Interestingly, integrated transcriptomic and metabolic analysis uncovered GFPT1-driven Uridine Diphosphate-N-Acetylglucosamine (UDP-GlcNAc) accumulation and increased taurine levels. Validation experiments confirmed *GFPT1* upregulation and *ANGPTL4* downregulation through RT-qPCR and increased O-GlcNAcylation via Western blot. Importantly, clinical datasets further supported the correlations between *GFPT1*/*ANGPTL4* expression and cholesterol levels in Non-Alcoholic Steatohepatitis (NASH) liver cancer patients. This work establishes a chronological paradigm of cholesterol sensing and identifies *GFPT1* and *ANGPTL4* as key regulators bridging glycosylation and lipid pathways, providing mechanistic insights into cholesterol-associated metabolic disorders.

## 1. Introduction

Cholesterol is a vital sterol molecule that serves as a fundamental structural component of eukaryotic cell membranes, where it modulates fluidity, permeability, and the formation of lipid rafts critical for cellular signaling [1,2]. Beyond its role in membrane integrity, cholesterol acts as a precursor for steroid hormones (e.g., cortisol, estrogen, and testosterone), bile acids, and vitamin D, making it indispensable for systemic metabolism [3]. Additionally, recent studies have highlighted its involvement in immune cell function and synaptic plasticity, further broadening its physiological significance [4,5]. Given its physiological importance, cholesterol homeostasis is tightly regulated through a balance of endogenous biosynthesis (via the mevalonate pathway), dietary uptake (mediated by LDL receptors), and efflux (via ABC transporters such as ABCA1 and ABCG1) [6,7]. Notably, post-translational modifications (e.g., ubiquitination of LDLR) and epigenetic regulation (e.g., DNA methylation of cholesterol synthesis genes) add further layers of complexity to this regulatory network [8,9]. Disruptions in these regulatory mechanisms contribute to severe pathological conditions, including atherosclerosis, coronary artery disease, and neurodegenerative disorders such as Alzheimer’s disease [10,11].

The global burden of cholesterol-related diseases is staggering. Cardiovascular diseases (CVDs), driven largely by hypercholesterolemia, account for nearly 18 million deaths annually, representing the leading cause of mortality worldwide [12,13]. Alarmingly, recent epidemiological data indicate a rising prevalence of dyslipidemia in younger populations, likely due to sedentary lifestyles and poor dietary habits [14]. Moreover, emerging evidence suggests that intracellular cholesterol accumulation in neurons and glial cells exacerbates neuroinflammation and amyloid-β aggregation, linking dysregulated cholesterol metabolism to the progression of Alzheimer’s disease [15,16]. Despite the widespread use of statins and other lipid-lowering therapies, a significant proportion of patients exhibit inadequate LDL-cholesterol control or suffer from drug-related side effects, underscoring the need for novel therapeutic strategies [17,18]. Emerging approaches, such as PCSK9 inhibitors and RNA-based therapies, show promise but face challenges in accessibility and long-term safety [19].

Despite decades of research, critical gaps remain in our understanding of cholesterol metabolism [20,21]. First, most studies have focused on isolated pathways (e.g., Sterol Regulatory Element-Binding Protein 2 (SREBP2)-mediated transcriptional regulation or LDLR-dependent uptake) rather than the dynamic, system-wide interactions governing cholesterol flux [22]. Second, conventional single-omics approaches (e.g., transcriptomics or metabolomics alone) fail to capture the intricate crosstalk between gene regulatory networks and metabolic remodeling, limiting mechanistic insights [23]. For example, while transcriptomic analyses can identify differentially expressed genes in cholesterol-fed models, they do not reveal how these changes translate into functional metabolic shifts. Conversely, metabolomic profiling detects metabolic alterations but lacks the resolution to pinpoint upstream genetic regulators.

To bridge these gaps, we implemented an integrated multi-omics strategy combining RNA sequencing (RNA-seq) and untargeted lipidomics to systematically dissect the genetic and metabolic networks controlling cholesterol homeostasis in mammalian cells. HeLa cells were selected as a model due to their well-characterized cholesterol metabolism pathways, extensive use in lipid homeostasis studies, and relevance for exploring conserved cellular responses to sterol stress [24,25,26]. These cells retain core sterol-sensing machinery and have been validated for studying cholesterol-induced transcriptional and metabolic reprogramming, making them suitable for our integrated omics analysis. Our approach leverages advanced bioinformatic tools to correlate transcriptional changes with metabolic flux, enabling the identification of novel regulatory hubs and potential therapeutic targets. Importantly, this study not only expands the mechanistic understanding of cholesterol metabolism but also establishes a scalable framework for investigating other complex metabolic disorders. Given the rising prevalence of cardiovascular and neurodegenerative diseases, our findings hold significant translational potential for precision medicine and drug development.

## 2. Results

### 2.1. Transcriptome Analysis of HeLa Cells Stressed with Cholesterol

#### 2.1.1. Quality of Sequencing Data and Comparative Analysis

We first carried out a comprehensive quality control and preprocessing of our original sequencing data. The details were summarized in Appendix A. Transcriptome analysis revealed consistent gene detection across 12 samples (18,738–19,436 genes per sample), demonstrating robust technical reproducibility (Appendix A). Furthermore, the Log2 FPKM values exhibited a broad distribution, highlighting substantial heterogeneity in gene expression profiles (Appendix A). These findings establish a high-quality dataset for identifying candidate genes involved in cholesterol response.

To investigate the transcriptional effects of cholesterol treatment, we analyzed gene expression patterns in HeLa cells under vehicle (V, the same concentration of MβCD used to dissolve cholesterol) or 50 μM cholesterol-treated (C) conditions at 6 and 24 h. The sample correlation analysis revealed high Pearson coefficients (0.985–0.995) among biological replicates within the same treatment group, demonstrating excellent experimental reproducibility (Appendix A). Principal Component Analysis (PCA) further showed clear separation among treatment groups in two-dimensional space: C_6h and V_6h samples formed distinct clusters, while the 24 h groups (C_24h and V_24h) were differentiated from the 6 h groups along principal component axes (Appendix A). Notably, our observations that the V_6 and V_24 samples separate specifically along PC1 may be attributed to the use of cyclodextrin as a solvent at the 6 and 24 h timepoints; no significant difference was detected in PC2. In contrast, for cholesterol treatment, differences were observed not only in PC1 but also in PC2 at both the 6 and 24 h timepoints. These results indicate that both cholesterol treatment and time factors collectively drive specific transcriptional changes in HeLa cells.

#### 2.1.2. Identification and Functional Enrichment Analysis of DEGs

To comprehensively identify cholesterol-responsive genes, including those with subtle but significant expression changes, we defined differentially expressed genes (DEGs) using a |log2FC| > 0.38 (1.3-fold) and *p*-value < 0.05. This threshold was selected to prioritize sensitivity for moderate but statistically robust transcriptional shifts, which are characteristic of metabolic regulators and may be missed by conventional 2-fold cutoffs. Strikingly, the 6 h treatment elicited more pronounced changes, with 1340 genes upregulated and 976 downregulated compared to vehicle-treated cells, whereas the 24 h response showed moderate alterations (758 up/734 down). Key cholesterol regulators (*ABCA1*, *ABCG1*, *MYLIP*) were drastically induced at 6 h, while newly identified factors (*ANGPTL4*, *PTGS2*) emerged as potential mediators. The significant alterations in gene expression observed at the 6 h time point indicate that cells have initiated a rapid adaptive response. These early-responsive genes may play roles in modulating cholesterol metabolism and transport mechanisms to counteract the abrupt elevation in cholesterol levels. For example, upregulated cholesterol transport protein genes (such as *ABCA1* and *ABCG1*) facilitate the efflux of excess cellular cholesterol, thereby contributing to the maintenance of intracellular cholesterol homeostasis (Figure 1A,B, Appendix A). The relatively moderate changes in gene expression observed at 24 h may be associated with long-term regulatory mechanisms within the cell. At this stage, the cell may fine-tune its initial response or activate processes involved in repair and the maintenance of normal cellular functions in order to adapt to the persistently elevated cholesterol environment (Figure 1A,B, Appendix A). This temporal pattern suggests that the cellular response to cholesterol is stage-specific, with distinct sets of genes playing unique roles at different phases of cholesterol processing. To delineate temporal dynamics of cholesterol-mediated gene regulation, we conducted comparative Venn analysis of DEGs following 6 h versus 24 h cholesterol exposure (Figure 1C,D). Venn analysis demonstrated time-dependent transcriptional changes in cholesterol-treated HeLa cells. C_6h uniquely upregulated 1134 genes and downregulated 801 genes, while C_24h specifically induced 552 up- and 559 downregulated genes, with 206 up- and 175 downregulated genes shared between both timepoints, which may represent both timepoint-specific effectors and putative sustained cholesterol responders (core DEGs). Specifically, the heatmap results showed that in the 6 h cholesterol-treated group, *ABCA1*, *ABCG1*, and *MYLIP* were among the most significantly upregulated genes, while *ANGPTL4*, *HMGCS1*, and *MSMO1* were among the most downregulated ones (Figure 1E). In the 24 h treatment group, *HLA-B*, *PTGS2*, *CXCL8*, etc., were the most upregulated, whereas *OLAH*, *HS3ST5*, and *RASGRP4* were the most downregulated (Figure 1F). Together, cellular responses to cholesterol are stage-specific, with distinct genetic mechanisms playing unique roles at different developmental stages, thereby exhibiting temporal dynamic characteristics.

#### 2.1.3. Gene Enrichment Analysis Reveals Functional Pathway Differences in HeLa Cells Responding to Cholesterol Stimulation

To further elucidate the time-dependent effect of cholesterol, we conducted GOBP (Gene Ontology Biological Process) and KEGG (Kyoto Encyclopedia of Genes and Genomes) enrichment analyses on the differentially expressed genes. The results revealed the following: For C_6h_up genes, GOBP enrichment focused on “response to nutrient levels” and “cellular stress response”, including endoplasmic reticulum stress-related functions such as “response to unfolded protein”. KEGG pathway analysis showed activation of the “TNF signaling pathway” and “FoxO signaling pathway”, suggesting short-term cholesterol stimulation preferentially triggers cellular stress and immune surveillance signals (Figure 2A). Conversely, C_6h_down genes were significantly enriched in lipid synthesis-related GOBP terms (e.g., “sterol biosynthetic process”, “cholesterol metabolic process”), with KEGG pathways dominated by “steroid biosynthesis” and “terpenoid backbone biosynthesis”—indicating 6 h treatment directly suppresses de novo sterol synthesis in cells (Figure 2B).

In the 24 h treatment group, GOBP enrichment of C_24h_up genes extended to functions like “cellular response to external stimulus”, “response to starvation”, and “fat cell differentiation”. KEGG pathways showed activated immune and inflammation-related cascades, including the “NF-κB signaling pathway” and “MAPK signaling pathway”—reflecting the increased complexity of cellular stress responses and sustained immune pathway activation under prolonged stimulation (Figure 2C). For C_24h_down genes, although “sterol biosynthetic process” remained a key target, KEGG analysis also revealed significant enrichment of metabolic regulatory pathways (e.g., “PPAR signaling pathway”, “fatty acid degradation”), implying 24 h treatment may coordinate lipid metabolism homeostasis via multi-pathway regulation beyond direct sterol synthesis inhibition (Figure 2D).

To systematically characterize pathway-level responses to cholesterol treatment, we performed Gene Set Enrichment Analysis (GSEA) using the KEGG database in HeLa cells following 6 h and 24 h cholesterol exposure (Appendix A). The results demonstrated that 6 h treatment significantly activated ABC transporters (NES = 1.98, FDR = 0.008) while suppressing steroid biosynthesis (NES = −2.38, FDR = 9.66 × 10^−8^). In contrast, 24 h treatment specifically enriched cytokine–cytokine receptor interaction (NES = 1.85, FDR = 0.0004) and NOD-like receptor signaling (NES = 2.07, FDR = 0.001). Strikingly, terpenoid backbone biosynthesis was consistently downregulated at both timepoints (6 h NES = −2.18; 24 h NES = −1.93), suggesting that cholesterol orchestrates temporal coordination between metabolic reprogramming and inflammatory responses.

Our transcriptomic analysis revealed that cholesterol treatment induced early (6 h) upregulation of TNF alongside FoxO signaling pathway genes, while NF-κB targets were enriched at 24 h. The delayed NF-κB activation (24 h) coincided with secondary TNF upregulation, consistent with TNF-driven NF-κB signaling. These results support a two-phase mechanism: (1) cholesterol-induced metabolic stress (ROS/ERS) activates FoxO to initiate TNF transcription, followed by (2) TNF-dependent NF-κB amplification, establishing a pro-inflammatory feed-forward loop [27,28]. Together, these findings indicate that cholesterol treatment elicits time-dependent reprogramming of gene functions in HeLa cells: short-term exposure prioritizes “stress surveillance+direct sterol synthesis inhibition”, while long-term exposure evolves into “complex stress responses+immune activation+multi-pathway metabolic regulation”. This dynamic shift provides hierarchical, time-resolved functional and pathway insights to guide the identification of cholesterol-sensing genes and establishes a foundation for dissecting the molecular temporal mechanisms governing cholesterol homeostasis.

#### 2.1.4. Weighted Gene Co-Correlation Network Analysis (WGCNA)

To decipher the dynamic rewiring of gene co-expression networks under cholesterol treatment, we performed WGCNA analysis on HeLa cell transcriptomes at the 6 h and 24 h timepoints. WGCNA clusters genes with similar expression patterns into “modules,” each labeled with a distinct color (e.g., brown, yellow), where genes within the same module are likely involved in coordinated biological processes. First, we validated the reliability of our network construction. The sample dendrogram (Appendix A) demonstrated robust clustering of biological replicates, ensuring analytical reliability. Scale-free topology validation (Appendix A) confirmed appropriate soft-thresholding (R^2^ > 0.9). Module clustering (Figure 3A) and an eigengene heatmap (Appendix A) indicated high module independence. Next, we proceeded to identify modules associated with cholesterol treatment. Module–trait correlation analysis (Figure 3B) revealed two critical modules: the brown module was significantly inhibited (r = −0.87, *p* = 2 × 10^−4^) after 6 h of cholesterol treatment, while it showed an inducing trend (r = 0.65, *p* = 0.02) after 24 h of control treatment. This biphasic regulatory pattern suggests that it may play a time-dependent regulatory role in the dynamic balance of cholesterol. In contrast, the yellow module was significantly activated (r = 0.73, *p* = 0.007) after 6 h of cholesterol treatment and inhibited (r = −0.7, *p* = 0.01) after 6 h of vehicle treatment. This early, rapid, and treatment-specific response feature is highly consistent with its potential role as a cholesterol sensor. The differential regulatory patterns of the two modules provide important clues for understanding the molecular mechanism of cholesterol perception and metabolic regulation (Figure 3B). We selected the yellow and brown modules of WGCNA for further analysis. Taking the intersection with the common upregulated and downregulated parts of the previously identified differentially expressed genes, 190 genes were obtained for protein–protein interaction (PPI) (Figure 3C, Appendix A). PPI analysis of hub genes from brown and yellow WGCNA modules via the STRING database (https://string-db.org) (accessed on 12 June 2025) (Figure 3D) identified multiple metabolism-associated candidates, *HMGCR* (a key enzyme in sterol biosynthesis) and *XBP1* (a mediator of endoplasmic reticulum stress) linked the modules to cholesterol homeostasis and stress responses. Strikingly, the network included *ANGPTL4* (angiopoietin-like 4) and *GFPT1* (glutamine-fructose-6-phosphate transaminase 1), both implicated in cholesterol metabolism and insulin signaling. *ANGPTL4* connections align with its known role in lipid metabolism via LPL inhibition [29], while *GFPT1* linkages to hexosamine pathway components suggest cholesterol-mediated glycosylation regulation [30]. Their topological prominence underscores their potential as cholesterol-sensing regulators.

### 2.2. Metabolome Analyses

#### 2.2.1. Lipidomics Analysis

To comprehensively investigate the metabolic changes induced by cholesterol treatment in HeLa cells, we performed untargeted metabolomics analysis using liquid chromatography mass spectrometry (LC-MS). PCA analysis of the lipidomic data showed that, in the negative ion mode, highly overlapping lipidomic profiles existed between the C_6h and V_6h groups (PC1: 52% explained variance), revealing consistent metabolic patterns between conditions (Appendix A). However, consistent separation was observed in the positive ion mode (PC1: 46% variance) (Appendix A). Our findings demonstrate that a 6 h cholesterol exposure is sufficient to remodel the lipidome of HeLa cells.

Similarly, in order to detect the significant changes in metabolites to the greatest extent, we defined differentially accumulated metabolites (DAMs) based on a |log2FC| > 0.38 (1.3-fold) and a *p*-value < 0.05. Negative ion mode measurements revealed significant regulation of 75 lipid species (25 upregulated, 50 downregulated) (Figure 4A, Appendix A), while the positive ion mode detected more extensive remodeling, with 1176 differentially abundant lipids (373 increased, 803 decreased) (Figure 4B, Appendix A). This robust lipidomic reprogramming suggests that cholesterol rapidly orchestrates complex metabolic shifts, potentially through selective modulation of specific lipid pathways.

The lipidomic analysis revealed significant alterations in lipid species between the cholesterol- and vehicle-treated groups, highlighting distinct modulation patterns in both the positive and negative ion modes. In the negative ion mode, the levels of multiple phosphatidylserine (PS) species (such as PS (22:6/0:0) (2-fold, *p* < 0.05) and PS (22:4/0:0) (0.3-fold, *p* < 0.05)) in the cholesterol treatment group were significantly altered, suggesting that cholesterol may have a broad impact on the PS remodeling process of the cell membrane. This phenomenon may be related to membrane dynamics, lipid metabolic reprogramming, or the regulation of cell signal transduction. Cholesterol treatment significantly upregulated gangliosides GM1 (d18:0/16:0; 1.9-fold, *p* < 0.05) and GM2 (d18:1/18:0; 1.8-fold, *p* < 0.05), as well as ether-linked phospholipid PE-NMe2 (1.8-fold, *p* < 0.05). These coordinated increases suggest enhanced glycosphingolipid metabolism and potential modulation of neuronal function. Unexpectedly, trace levels of temelastine and thifluzamide (environmental contaminants or media components) were detected, but their biological relevance remains unclear (Figure 4C). In the positive mode, triglycerides (TGs) such as TG (17:0/22:6/22:6) were notably upregulated in the cholesterol group, suggesting enhanced lipid storage or metabolism. Cholesterol treatment markedly upregulated lysophosphatidylcholines (LysoPCs) and phosphatidylcholines (PCs), including LysoPC (20:4) (5.2-fold, *p* < 0.01), PC (18:4/0:0) (8.3-fold, *p* < 0.05), and PC (22:6/0:0) (6.3-fold, *p* < 0.01). These pronounced increases suggest active membrane phospholipid remodeling and potential alterations in lipid-mediated signaling pathways. Additionally, unique metabolites such as 3-deoxyvitamin D3 derivatives and methylmercury were detected, possibly reflecting oxidative stress or exogenous exposures (Figure 4D).

Negative ion mode KEGG enrichment analysis identified the glutathione metabolism pathway within sulfur metabolism as highly significant (Figure 4E). In the positive ion mode, KEGG pathway analysis demonstrated statistically significant enrichment of the mTOR signaling pathway and Salmonella infection, along with elevated pathway activity in Chagas disease (American trypanosomiasis) (Figure 4F). These dual-mode data underscore how infection stress couples the sulfur-redox system (GSH) and energy-sensing (mTOR) with lipid metabolism remodeling. Collectively, these findings underscore a broad reorganization of lipid classes under cholesterol modulation, with perturbations in glycerolipids, phospholipids, and sphingolipids. The data also suggest roles for these lipids in membrane dynamics, signaling, and metabolic stress under cholesterol treatment, providing a foundation for further mechanistic studies.

#### 2.2.2. Broad-Spectrum Metabolomics Analysis

For broad-spectrum metabolomics analysis, the results demonstrated that in the negative ion mode, PC1 and PC2 explained 47% and 28% of the variance, respectively, with a distinct separation between the control group (V_6h) and cholesterol-treated group (C_6h) in the PCA score plot (Appendix A). In the positive ion mode, PC1 and PC2 accounted for 49% and 31% of the variance, respectively, and the two groups showed a remarkable distinction (Appendix A). These results suggest that a 6 h cholesterol treatment leads to significant alterations in the water-soluble metabolic profile of HeLa cells.

Volcano plots of the negative ion mode measurements revealed significant changes in 172 metabolites (82 upregulated, 90 downregulated) (Figure 5A, Appendix A), while positive ion mode detected 244 differentially abundant metabolites (146 increased, 98 decreased) (Figure 5B, Appendix A). This extensive metabolic reprogramming indicates that cholesterol rapidly modulates diverse biochemical pathways, with polarity-dependent variations in metabolic responses.

The metabolomic profiling identified significant alterations in metabolite levels between the cholesterol and vehicle control groups, highlighting distinct metabolic pathway perturbations. In the negative ion mode, key metabolites such as O-phosphorylethanolamine (a precursor for phospholipid synthesis) and 1-oleoyl-lysophosphatidic acid (a signaling lipid involved in inflammation and fibrosis) were significantly upregulated in the cholesterol group, suggesting cholesterol may promote the accumulation of phospholipid precursors or the activation of inflammatory signals [31,32]. Additionally, adenine, a fundamental component of nucleic acids and energy carriers like ATP, showed altered levels, suggesting potential disruptions in nucleotide metabolism or cellular energy dynamics under cholesterol stress. These findings underscore the impact of cholesterol on critical metabolic pathways, including phospholipid biosynthesis and inflammatory signaling. The elevation of D-(−)-glutamine, a central player in nitrogen metabolism, could signify increased demand for amino acid metabolism and potential activation of glutaminolysis pathways (Figure 5C). This response may also reflect cellular adaptation to oxidative stress, as glutamine serves as a precursor for the synthesis of glutathione (GSH), a critical antioxidant. In the positive ion mode, choline and O-acetylcholine (critical for neurotransmission and membrane integrity) were markedly altered, implying cholesterol-induced effects on cholinergic signaling. Notably, palmitoyl sphingomyelin (a major sphingolipid in membranes) and C-8 ceramide-1-phosphate (a bioactive lipid mediator) showed differential expression, pointing to sphingolipid metabolism dysregulation. Furthermore, the difference in the expression of 7-aminomethyl-7-deaminoguanine (a modified purine base analogue) is also worthy of attention. This change may reflect the abnormalities in nucleic acid metabolism or epigenetic modifications under cholesterol intervention (Figure 5D).

KEGG pathway enrichment analysis in negative ion mode revealed significant enrichment of lipid-related pathways, including the sphingolipid signaling pathway, sphingolipid metabolism, and glycerophospholipid metabolism, along with notable changes in amino acid metabolism pathways such as taurine and hypotaurine metabolism (Figure 5E). Positive ion mode analysis demonstrated significant enrichment of primary bile acid biosynthesis, glycerophospholipid metabolism, and glutathione metabolism, with primary bile acid biosynthesis being directly related to cholesterol metabolism (Figure 5F). These findings suggest that lipid metabolic reprogramming and bile acid biosynthesis may play crucial roles in cholesterol sensing in HeLa cells, along with potential implications for oxidative stress. Collectively, these findings demonstrate that cholesterol intervention leads to broad metabolic reprogramming, affecting pathways such as phospholipid synthesis, energy metabolism, neurotransmission, and sphingolipid signaling. The observed shifts may reflect adaptive responses to cholesterol overload or compensatory mechanisms to maintain cellular homeostasis. Further functional studies are warranted to elucidate the mechanistic links between these metabolic changes and cholesterol-associated physiological or pathological outcomes.

### 2.3. Integration of Transcriptome and Metabolome Profiles

To elucidate the molecular network underlying cholesterol-mediated metabolic reprogramming, we integrated transcriptomic and metabolomic data through correlation-based network analysis and joint pathway enrichment. The integrative analysis uncovered significant crosstalk between transcriptional regulation and metabolic remodeling in cholesterol-treated HeLa cells. Our analysis identified the TNF signaling pathway (*p* = 1.08 × 10^−6^), insulin resistance (*p* = 4.91 × 10^−5^), and Hippo signaling pathway (*p* = 9.80 × 10^−5^) as the most significantly enriched pathways, exhibiting notable impact values (e.g., 0.68 for insulin resistance) (Figure 6A, Appendix A). Furthermore, key lipid-associated pathways, including glycerophospholipid metabolism (impact = 1.64) and steroid biosynthesis (impact = 1.61), demonstrated pronounced alterations (Figure 6B). These observations imply that cholesterol exposure likely disrupts cellular homeostasis by dysregulating inflammatory responses, metabolic balance, and lipid dynamics.

At the molecular level, several key findings emerged. First, the elevated expression of *GFPT1* (1.8-fold change) and subsequent accumulation of UDP-GlcNAc (2.2-fold change) indicated enhanced protein O-GlcNAcylation—a critical post-translational modification involved in nutrient sensing, insulin resistance, and inflammatory responses (Figure 6C, Appendix A). Second, cholesterol treatment induced coordinated changes in phospholipid metabolism and choline homeostasis. Transcriptomic analysis showed upregulation of *PLD1* (phospholipase D1) and downregulation of the choline transporter *FLVCR1* (choline and heme transporter 1), accompanied by metabolomic evidence of reduced choline levels. Upregulated PC species, including polyunsaturated and lysophosphatidylcholine forms, suggest enhanced PC remodeling, while downregulated ether lipids and plasmalogens indicate disrupted membrane lipid synthesis. The PLD1-mediated PC catabolism, combined with impaired choline reuptake via FLVCR1, likely disrupts the CDP-choline pathway, driving choline depletion and phospholipid remodeling. These changes may alter membrane rigidity and promote pro-oxidative stress responses, linking cholesterol-induced metabolic shifts to cellular dysfunction. Additionally, our data showed a marked elevation in intracellular taurine levels accompanied by increased expression of the taurine transporter gene *SLC6A6* upon cholesterol treatment. This suggests that cholesterol promotes taurine accumulation via *SLC6A6* upregulation, potentially as a compensatory mechanism to mitigate cholesterol-induced oxidative stress. Given taurine’s well-established role as an antioxidant and membrane stabilizer, its accumulation may represent an adaptive cellular response to maintain redox homeostasis and counteract oxidative stress triggered by cholesterol overload. These findings highlight the *SLC6A6*-taurine pathway as a novel modulator of cholesterol homeostasis and oxidative stress resilience.

Notably, cholesterol exposure significantly upregulated both octacosyl triacontanoate levels and *ABCD1* transcription, with a strong positive correlation between this lipid species and *ABCD1* expression. Given *ABCD1*’s established role as a peroxisomal Very-Long-Chain Fatty Acids (VLCFA) transporter, these observations suggest a potential link between cholesterol signaling and VLCFA metabolism. The coordinated upregulation of octacosyl triacontanoate (a VLCFA-derived ester) and ABCD1 raises the possibility that cholesterol may promote peroxisomal handling of VLCFAs, possibly through ABCD1-mediated transport for subsequent β-oxidation. These findings highlight a previously unrecognized connection between cholesterol homeostasis and VLCFA ester metabolism, warranting further investigation into ABCD1’s functional involvement in this process.

Furthermore, cholesterol treatment significantly increased the level of sphingomyelin SM(d18:2/16:0) while downregulating the lipid raft marker gene *CAV1*. This inverse correlation suggests that cholesterol may disrupt *CAV1*-mediated lipid raft organization, leading to SM(d18:2/16:0) accumulation in the plasma membrane and potentially altering membrane dynamics or signaling efficiency. Our integrated transcriptomic and metabolomic analysis revealed that cholesterol treatment induced a significant depletion of cholesterol ester CE (22:2 (13Z, 16Z)) concomitant with upregulation of cholesterol acyltransferases *SOAT1*, suggesting a substrate-specific diversion of polyunsaturated fatty acids from esterification pathways during cholesterol overload. This metabolic shift correlated with strong coordinated expression changes (either induction or repression) in a 16-gene network (selected by Pearson correlation analysis with CE (22:2) levels), encompassing membrane regulators (*NT5E*, *TSPAN9*), extracellular matrix components (*THBS1*, *HTRA3*), transcriptional controllers (*LHX2*, *MYOCD*), and DNA damage responders (*NEIL2*, *CCM2*), forming a comprehensive adaptive network (Appendix A).

Particularly noteworthy were the robust inductions of *THBS1* and *MYOCD*, known mediators of vascular remodeling, along with *NT5E*, which may couple purinergic signaling with lipid metabolism. These findings collectively suggest that cholesterol orchestrates a multifaceted response involving selective cholesterol ester remodeling, membrane homeostasis maintenance, and activation of stress-adaptive pathways, with the identified gene cluster representing promising targets for modulating cholesterol-related metabolic dysregulation.

Together, these results demonstrate a tightly regulated molecular network in which transcriptional changes and metabolic adaptations work in concert to modulate cholesterol trafficking, storage, and utilization in response to excess cholesterol. The integration of multi-omics approaches has provided novel insights into the complex interplay between cholesterol metabolism and broader cellular regulatory networks.

### 2.4. Validation

To validate the reliability of our transcriptomic and metabolomic findings, we prioritized key targets based on their network centrality and pathway relevance. *GFPT1* and *ANGPTL4* were selected for validation due to their central roles in linking glycosylation (*GFPT1*) and lipid metabolism (*ANGPTL4*) in our integrated network analysis (Figure 6C and Figure 3D), while O-GlcNAcylation was prioritized as a key post-translational modification downstream of GFPT1-mediated UDP-GlcNAc accumulation. Quantitative real-time PCR (qRT-PCR) validated the RNA-seq results (primer sequences are listed in Appendix A), showing consistent upregulation of *GFPT1* (RNA-seq: 1.8-fold; qPCR: 3.4-fold, *p* < 0.01) and consistent downregulation of *ANGPTL4* (RNA-seq: 0.08-fold; qPCR: 0.2-fold, *p* < 0.01) in 6 h cholesterol-treated cells (Figure 7A). Western blotting analysis of HeLa cells treated with cholesterol demonstrated a significant increase (1.5-fold, *p* < 0.001 with 50 μM cholesterol; 1.6-fold, *p* < 0.001 with 75 μM cholesterol; 1.1-fold, *p* < 0.05 with 100 μM cholesterol) in protein O-GlcNAcylation levels compared to vehicle-treated controls, which aligns with the GFPT1-driven UDP-GlcNAc accumulation observed in our metabolomics data. (Figure 7B). The consistency between qPCR and RNA-seq results, along with the upregulation of protein O-GlcNAcylation (a functional readout of GFPT1 activity), confirms the robustness of our omics-driven discoveries and supports a functional role for the GFPT1/UDP-GlcNAc/O-GlcNAcylation axis in cholesterol response.

Finally, we compared the changes of some differentially expressed genes in the diseases of NASH liver cancer patients in the GEO database (GSE126848), and the results were also verified (Figure 7C). These findings collectively indicate that cholesterol may regulate critical cellular processes through two distinct mechanisms: (1) transcriptional regulation of gene expression and (2) post-translational modification of proteins via glycosylation. The complementary nature of these omics approaches provides compelling evidence for cholesterol’s multifaceted role in cellular regulation, establishing a solid foundation for further investigation into cholesterol-mediated signaling pathways.

## 3. Discussion

Our integrated multi-omics study provides a comprehensive temporal map of cholesterol-induced cellular reprogramming in HeLa cells, revealing distinct yet interconnected phases of metabolic and transcriptional adaptation. The findings establish that cholesterol sensing occurs through a precisely orchestrated sequence of molecular events, beginning with rapid metabolic suppression and culminating in complex immunometabolic rewiring.

The early response phase (6 h) was characterized by two dominant features: induction of cholesterol efflux machinery (*ABCA1*/*ABCG1*) and simultaneous suppression of endogenous sterol synthesis (*HMGCS1*/*MSMO1* downregulation). This biphasic regulation likely represents a homeostatic attempt to reduce cellular cholesterol load while conserving energy by repressing redundant biosynthesis pathways [33,34]. Notably, the immediate activation of stress response pathways (unfolded protein response, TNF signaling) suggests cholesterol overload is perceived as both a metabolic and proteostatic challenge [35]. Transition to the late phase (24 h) revealed a striking shift toward inflammatory and immune-modulatory programs, evidenced by NF-κB/MAPK activation and cytokine signaling upregulation. This temporal progression implies that prolonged cholesterol exposure triggers defensive mechanisms beyond pure metabolic adjustment, potentially preparing cells for microenvironmental communication. The persistent suppression of terpenoid backbone biosynthesis across both timepoints underscores cholesterol’s role as a master regulator of isoprenoid flux [36].

Our metabolomic findings provide a crucial missing link between transcriptional changes and functional outcomes. The accumulation of very-long-chain fatty acid esters (e.g., octacosyl triacontanoate) coupled with *ABCD1* upregulation reveals a previously unrecognized peroxisomal adaptation to cholesterol stress [37]. Similarly, the GFPT1-UDP-GlcNAc-O-GlcNAcylation axis suggests O-linked protein glycosylation as a novel regulatory layer in cholesterol signaling [38]. These discoveries substantially expand the conventional view of cholesterol metabolism beyond membrane biology and steroidogenesis. Notably, the GFPT1-UDP-GlcNAc axis may directly link cholesterol overload to NASH progression, where excessive O-GlcNAcylation has been implicated in driving hepatic insulin resistance and inflammation [39,40]. This connection warrants further investigation given the clinical overlap between hypercholesterolemia and NAFLD/NASH comorbidities.

The WGCNA-identified ANGPTL4/*GFPT1* network offers a plausible mechanism for how cells respond to cholesterol stress by coordinating lipid handling with nutrient sensing. *ANGPTL4*’s known function in lipid partitioning, combined with *GFPT1*’s role in hexosamine signaling, positions these factors as potential nodes integrating cholesterol status with broader lipid- and glucose-associated metabolic programs. Their co-regulation in our system may explain clinical observations linking cholesterol dysregulation to insulin resistance. Several findings have particular pathophysiological relevance. First, the cholesterol-induced disruption of CAV1-mediated lipid rafts (evidenced by SM(d18:2/16:0) accumulation) provides a molecular basis for altered signaling fidelity in hypercholesterolemia. Second, the selective depletion of polyunsaturated CE (22:2) species despite *ACAT1*/*SOAT1* upregulation suggests a membrane quality control mechanism that may be compromised in metabolic diseases. Finally, the strong concordance with NASH patient gene expression patterns underscores the translational relevance of our in vitro model. While this study provides a detailed temporal atlas, certain limitations warrant consideration. The HeLa cell model, while informative, may not fully capture tissue-specific cholesterol handling. Future work should explore whether the identified phases persist in primary cells and intact tissues. Additionally, the functional consequences of key observations (e.g., O-linked N-acetylglucosamine changes) require direct experimental validation.

In conclusion, our work establishes that cholesterol sensing is not a static event but a dynamic process evolving from acute metabolic correction to systemic stress adaptation. The identified temporal modules and molecular players—particularly the *ANGPTL4*, *GFPT1*, and peroxisomal lipid remodeling—offer new avenues for understanding cholesterol-related pathologies. These findings suggest that therapeutic strategies may need to take into account the temporal dimension of cholesterol dysregulation, potentially targeting different pathways in early versus chronic disease stages.

## 4. Materials and Methods

### 4.1. Chemicals

Trizol (Cat# 15596026) was purchased from Invitrogen (Carlsbad, CA, USA). MβCD (Cat# 332615) and Cholesterol (Cat# C3045) were purchased from Sigma-Aldrich (Saint Louis, MO, USA). Mouse monoclonal [RL2] to O-Linked N-Acetylglucosamine (Cat# ab2739) was purchased from Abcam (Cambridge, UK). GAPDH (Cat# 60004-1-Ig) was purchased from Proteintech (Wuhan, China).

### 4.2. Cell Culture

HeLa cell lines were acquired from ATCC and maintained in Dulbecco’s modified Eagle’s medium (DMEM, Gibco (Grand Island, NY, USA), Cat# 11965092)). The medium was supplemented with 10% fetal bovine serum (Gibco, Carlsbad, CA, USA) and 1% Penicillin/Streptomycin (Gibco, Carlsbad, CA, USA) solution, and cells were cultured in a humidified incubator at 37 °C with 5% CO_2_.

### 4.3. Western Blotting

Cells were first homogenized in ice-cold RIPA lysis buffer (50 mM Tris-HCl pH 7.4, 150 mM NaCl, 1% Triton X-100, 0.1% SDS, and 0.5% sodium deoxycholate), supplemented with a protease inhibitor mixture to prevent degradation. After centrifugation at 12,000× *g* for 15 min at 4 °C, the supernatant was collected, and protein concentration was determined using a BCA protein quantification kit (Beyotime Biotechnology, Shanghai, China) following the manufacturer’s protocol. Equal amounts of protein (30 µg) were then resolved via 10% SDS-polyacrylamide gel electrophoresis (SDS-PAGE) and electrotransferred onto a nitrocellulose membrane (Millipore, Billerica, MA, USA) at 200 mA for 120 min. The membrane was blocked with 5% milk-TBST at room temperature for 1 h, primary antibodies (mouse monoclonal anti-O-GlcNAc, Abcam #ab2739, 1:2000) (mouse monoclonal anti-GAPDH, Proteintech #60004-1-Ig, 1:10,000) were incubated overnight at 4 °C, followed by incubation with HRP-conjugated affinipure goat anti-mouse secondary antibodies (1:5000, Proteintech #SA00001-1) for 1 h at room temperature. Enhanced chemiluminescence solution was used to detect signals for visualizing protein expression.

### 4.4. Transcriptome Analysis

Following treatment with cholesterol for the indicated period, total RNA was extracted from cells with TRIzol reagent. RNA sequencing libraries were constructed using 1 µg of total RNA per sample with the NEBNext^®^ UltraTM RNA Library Prep Kit for Illumina^®^ (New England Biolabs, Ipswich, MA, USA) according to the manufacturer’s protocol. Briefly, mRNA was first isolated from total RNA using poly-T oligo-attached magnetic beads. The purified mRNA was fragmented at elevated temperature in the presence of divalent cations in NEBNext First Strand Synthesis Reaction Buffer (5×). First-strand cDNA was synthesized using random hexamer primers and M-MuLV Reverse Transcriptase (RNase H), followed by second-strand cDNA synthesis with DNA Polymerase I and RNase H. The cDNA fragments were then blunt-ended via exonuclease/polymerase activities, adenylated at the 3′ ends, and ligated to NEBNext Adaptors with hairpin loop structures. cDNA fragments of 250–300 bp were size-selected using the AMPure XP system (Beckman Coulter, Beverly, MA, USA). The size-selected, adaptor-ligated cDNA was treated with 3 µL of USER Enzyme (NEB, USA) at 37 °C for 15 min, followed by heat inactivation at 95 °C for 5 min. PCR amplification was performed using Phusion High-Fidelity DNA Polymerase, universal PCR primers, and index (X) primers. Final libraries were purified using the AMPure XP system and assessed for quality on an Agilent Bioanalyzer 2100 system (Agilent Technologies, Santa Clara, CA, USA). Index codes were incorporated during PCR to enable multiplexed sequencing.

The transcriptome analysis of HeLa cells included 12 samples divided into four groups (3 samples per group), with each group treated with vehicle (V) or cholesterol (C) for 6 h and 24 h, respectively. After screening and checking for sequencing errors, the raw sequencing data resulted in a total of 81.09 GB of clean data. The average clean data for the samples was 6.76 GB. The proportion of Q20 in each mRNA sample was above 98%, the proportion of Q30 was above 94%, and the GC content was between 48% and 51%, meeting the requirements for subsequent analysis. The data are shown in Appendix A. It can be observed that the outcome of the initial data preprocessing is highly satisfactory and fully capable of fulfilling the requirements for subsequent data processing and analysis.

### 4.5. RNA-Seq Data Validation Using qRT-PCR

Total RNA was extracted from cells with TRIzol reagent. cDNA was synthesized using HiScript II Q RT SuperMix (R223, Vazyme, Nanjing, Jiangsu, China), and DNA was removed by adding DNase. qRT-PCR was performed on a CFX-96 real-time PCR detection system (Bio-Rad, Hercules, CA, USA) with SYBR Green Master Mix (Sparkjade, Jinan, China). Relative gene expression data were analyzed using the 2^−ΔΔCT^ method. Primer sequences are listed in Appendix A.

### 4.6. Metabolome Analysis

#### 4.6.1. Metabolite Extraction

For lipidomics, 100 µL of each sample was transferred into a new 1.5 mL EP tube, followed by the addition of 300 µL of cold methanol. After vortex mixing, the samples were ground using a TissueLyser (with the frequency adjusted to 50 Hz for 5 min) and then incubated at −20 °C for 2 h. Subsequently, the samples were centrifuged at 25,000× *g* at 4 °C for 15 min, and 350 µL of the supernatant was transferred to a new EP tube and centrifuged again under the same conditions. Finally, 50 µL of supernatant from each sample was pooled to prepare quality control (QC) samples, and the remaining supernatant was transferred to 1.5 mL EP tubes for LC-MS analysis.

For broad-spectrum metabolomics, 25 mg of each tissue sample was weighed and mixed with 800 µL of precooled extraction reagent (methanol/acetonitrile/water = 2:2:1, *v*/*v*/*v*) containing internal standards (L-Methionine-d3, 4-Aminobutyric-2,2,3,3,4,4-d6 Acid, etc.). The mixture was homogenized for 5 min using a Tissue Lyser (JXFSTPRP, Shanghai, China), sonicated for 10 min, and incubated at −20 °C for 1 h. After centrifugation at 25,000 rpm for 15 min at 4 °C, the supernatant was collected and vacuum dried. The dried metabolites were resuspended in 600 µL of 70% acetonitrile, sonicated for 10 min at 4 °C, and centrifuged again at 25,000 rcf for 15 min. The final supernatant was transferred to autosampler vials for LC-MS/MS analysis.

#### 4.6.2. LC-MS/MS Analysis

Lipidomics analysis was performed using a 2777C UPLC system (Waters, Corporation in Milford, MA, USA) coupled with an Xevo G2-XS QTOF mass spectrometer (Waters, UK). Chromatographic separation was achieved on an ACQUITY UPLC CSH C18 column (100 mm × 2.1 mm, 1.7 μm, Waters, UK) maintained at 55 °C, with a flow rate of 0.4 mL/min. The mobile phase consisted of solvent A (ACN/H_2_O = 60:40, containing 0.1% formic acid and 10 mM ammonium formate) and solvent B (IPA/ACN = 90:10, containing 0.1% formic acid and 10 mM ammonium formate), using a gradient elution program: 0~2 min, 40–43% phase B; 2~7 min, 50–54% phase B; 7.1~13 min, 70–99% phase B; 13.1~15 min, 40% phase B, with an injection volume of 5 µL. For mass spectrometry, the Xevo G2-XS QTOF (Waters, Corporation in Milford, MA, USA) was operated in both the positive and negative ion modes: in the positive mode, capillary and sampling cone voltages were 3.0 kV and 40.0 V, respectively; in the negative mode, they were 2 kV and 40 V. Data were acquired in Centroid MSE mode, with a TOF mass range of 100 to 2000 Da (positive mode) and 50 to 2000 Da (negative mode), a survey scan time of 0.2 s, and MS/MS detection with all precursors fragmented using 19–45 eV and a scan time of 0.2 s. During acquisition, the LE signal was acquired every 3 s for mass accuracy calibration, and a quality control sample (pool of all samples) was acquired after every 10 samples to evaluate LC-MS stability.

Broad-spectrum metabolite separation and detection were conducted using a Waters 2D UPLC system (Waters, Corporation in Milford, MA, USA) coupled with a Q Exactive high-resolution mass spectrometer (Thermo Fisher Scientific, Waltham, MA, USA) equipped with a heated electrospray ionization (HESI) source. Chromatographic separation was achieved on an ACQUITY UPLC BEH Amide column (1.7 μm, 2.1 × 100 mm, Waters, USA) maintained at 30 °C. The mobile phase for the positive ion mode consisted of 0.1% formic acid and 10 mM ammonium formate in 95% acetonitrile (A) and 50% acetonitrile (B); for the negative ion mode, formic acid was omitted. The gradient elution program was as follows: 0–0.5 min, 2% B; 0.5–12 min, 2–50% B; 12–14 min, 98% B; 14–16 min, 98% B; 16–16.1 min, 98–2% B; and 16.1–18 min, 2% B, with a flow rate of 0.35 mL/min and an injection volume of 2 μL. Mass spectrometry parameters were set as follows: spray voltage of 3.8 kV (positive) and −3.2 kV (negative); sheath gas flow rate of 40 arb; auxiliary gas flow rate of 10 arb; auxiliary gas heater temperature of 350 °C; and capillary temperature of 320 °C. Full scans were performed in the range of 70–1050 *m*/*z* with a resolution of 70,000. The top 3 precursors were selected for MS/MS fragmentation with a resolution of 17,500 and stepped normalized collision energies of 15/30/45. Samples were analyzed in random order, with a QC sample inserted every 10 samples to minimize system errors.

#### 4.6.3. Bioinformatics Analysis

Bioinformatics analysis of the lipidomics data commenced with preprocessing of raw mass spectrometry data using Progenesis QI (version 2.2) for peak alignment, extraction, normalization, and deconvolution, followed by data correction via QC-RLSC. Statistical analyses included univariate methods such as *t*-test and fold change analysis, where *p*-values were subjected to FDR correction to obtain q-values, with differential ions screened based on thresholds of fold change ≥ 1.3 and *p*-value < 0.05. Differential ions were identified using Progenesis QI (version 2.2), and cluster analysis of these ions was conducted with log_2_-transformed data using the pheatmap function in R. Metabolite identification was based on searches against the LipidMaps database, with functional annotation and classification performed using the HMDB and KEGG databases, and metabolic pathway analysis of differential metabolites carried out using the KEGG database to determine involved biochemical and signal transduction pathways.

Bioinformatics analysis of the broad-spectrum metabolomics data commenced with the processing of raw LC-MS/MS data using Compound Discoverer 3.1 software, which involved peak extraction, retention time correction within and between groups, adduct ion pooling, missing value filling, background peak labeling, and metabolite identification by matching with BGI in-house library, mzCloud, and ChemSpider (integrating HMDB, KEGG, and LipidMaps) databases. The processed data were then imported into metaX for further preprocessing, including normalization via Probabilistic Quotient Normalization (PQN) to obtain relative peak areas, batch effect correction using QC-RLSC, and removal of compounds with a coefficient of variation (CV) greater than 30% in all QC samples. Univariate analyses, including fold change (FC) calculation and Student’s *t*-test, were used to screen differential metabolites with thresholds set as FC ≥ 1.3 and *p*-value < 0.05. Additionally, identified metabolites were classified and annotated using KEGG and HMDB databases, and metabolic pathway enrichment analysis based on KEGG was conducted to identify significantly enriched pathways (*p*-value < 0.05).

### 4.7. Statistical Analysis

Using GraphPad Prism 8.0 and Microsoft Excel, all statistical analyses were carried out. With the exception of specified instances, all data are presented as the mean ± SD. Employing a 95% confidence interval (CI), comparisons were tested via unpaired two-tailed Student’s *t*-test or one/two-way ANOVA with Tukey’s post hoc test. Statistical significance was defined as *p* ≤ 0.05.

## 5. Conclusions

This study systematically analyzed the regulatory network of cholesterol homeostasis in mammalian cells through the integration of transcriptomics and lipidomics multi-omics approaches. The research demonstrated that cholesterol processing could induce time-dependent transcriptional reprogramming in HeLa cells, initially activating stress response pathways while suppressing de novo sterol synthesis, and ultimately enhancing immune-inflammatory pathways and orchestrating multi-pathway metabolic regulation. Lipidomics analysis revealed that cholesterol exposure led to extensive remodeling of glycolipids, phospholipids, and sphingolipids, affecting key metabolic pathways such as sphingolipid metabolism and glycerophospholipid metabolism. Weighted gene co-expression network analysis combined with multi-omics integration identified hub genes, including ANGPTL4 and GFPT1, along with the associated signaling networks. These findings offer novel insights into the molecular mechanisms underlying cholesterol-related diseases and provide a foundation for the identification of potential therapeutic targets.

## Figures and Tables

**Figure 1 ijms-26-07108-f001:**
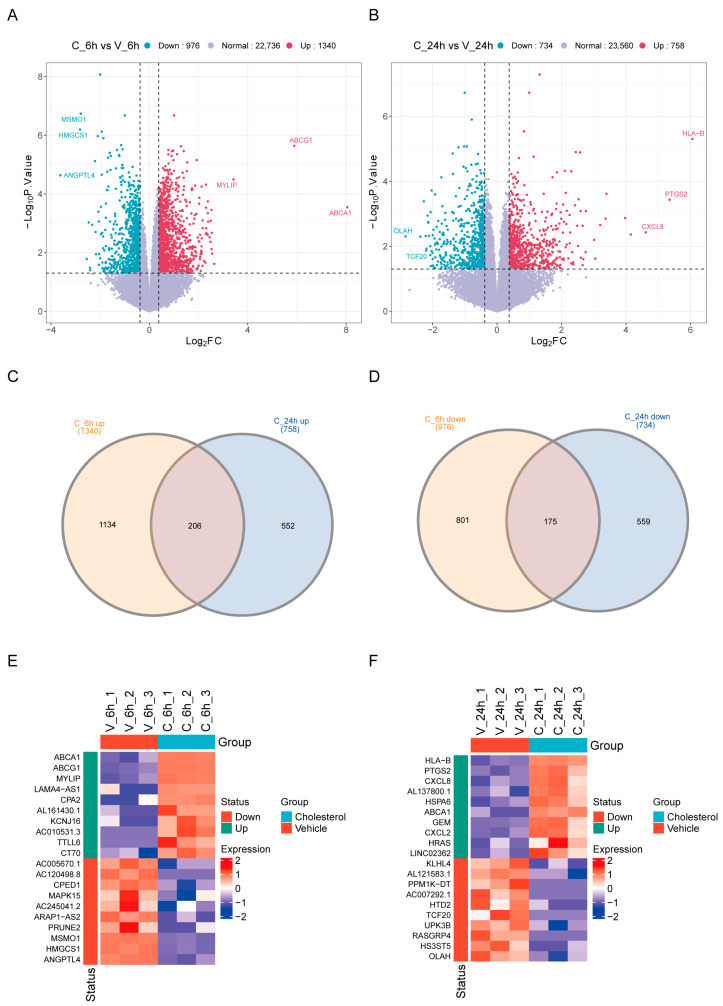
Differential expression analysis. (**A**,**B**) Volcano plots of DEGs in HeLa cells after 6 h (**A**: C_6h vs. V_6h) and 24 h (**B**: C_24h vs. V_24h) cholesterol exposure. DEGs are defined by |log2FC| > 0.38 (1.3-fold) and *p*-value < 0.05. Blue dots represent downregulated genes, red dots represent upregulated genes, and purple dots represent genes with no significant expression change. Numbers of downregulated, normal, and upregulated genes are indicated above each plot. (**C**,**D**) Venn diagram analysis of DEGs comparing 6 h and 24 h cholesterol treatments. (**E**,**F**) Heatmaps depicting the expression patterns of selected DEGs in HeLa cells. Panels illustrate gene expression changes between vehicle-treated and cholesterol-treated groups at 6 h and 24 h timepoints, respectively.

**Figure 2 ijms-26-07108-f002:**
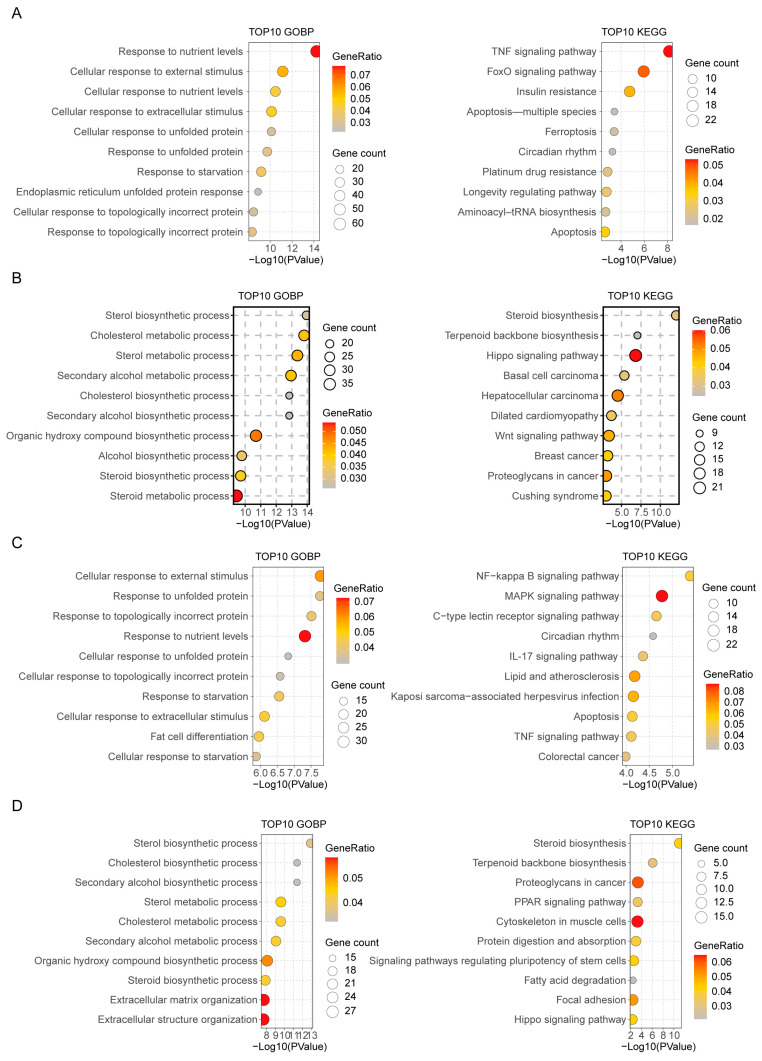
Gene enrichment analysis reveals functional pathway differences in HeLa cells responding to cholesterol stimulation. (**A**–**D**) GOBP and KEGG enrichment analyses of DEGs in HeLa cells after cholesterol stimulation. Panels show enrichment results for DEGs at different timepoints of cholesterol exposure, distinguishing biological processes and pathways.

**Figure 3 ijms-26-07108-f003:**
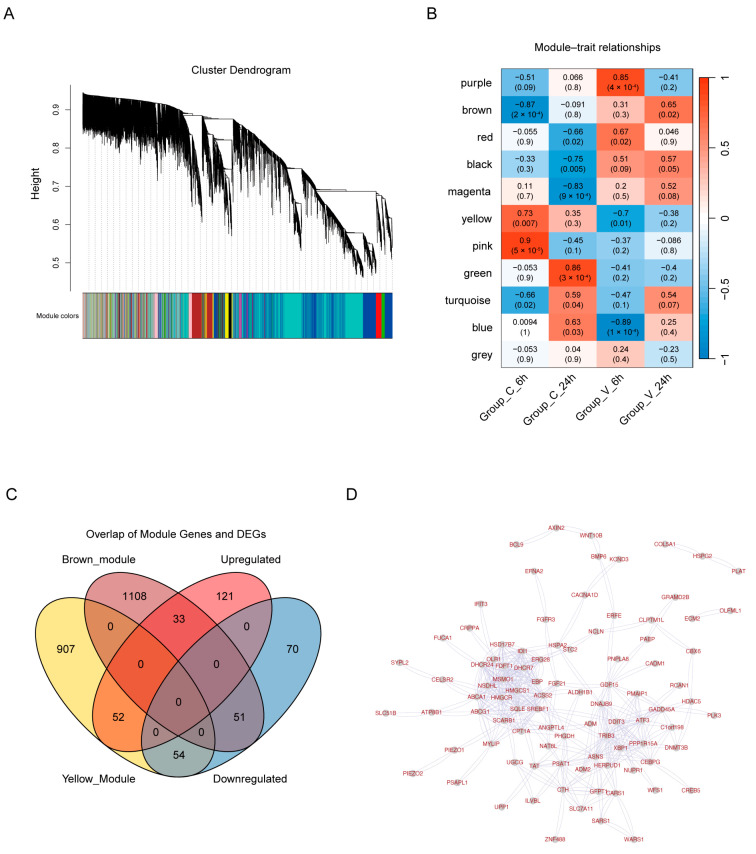
WGCNA of cholesterol-responsive gene networks. (**A**) Cluster dendrogram of genes. Vertical color bars represent co-expression modules identified by WGCNA, showing hierarchical clustering of genes based on expression similarity across samples. “Height” represents the degree of difference of genes or modules during the clustering process. (**B**) Heatmap depicting the correlation between WGCNA modules (rows) and experimental groups/traits. Corresponding *p*-values are in parentheses. The numbers outside the parentheses represent the correlation coefficients between the modules and the traits. (**C**) Venn diagram showing the intersection of genes from the brown and yellow WGCNA modules with up/downregulated DEGs. (**D**) STRING-based PPI network of hub genes from the brown and yellow WGCNA modules. Nodes represent proteins, and edges represent interactions.

**Figure 4 ijms-26-07108-f004:**
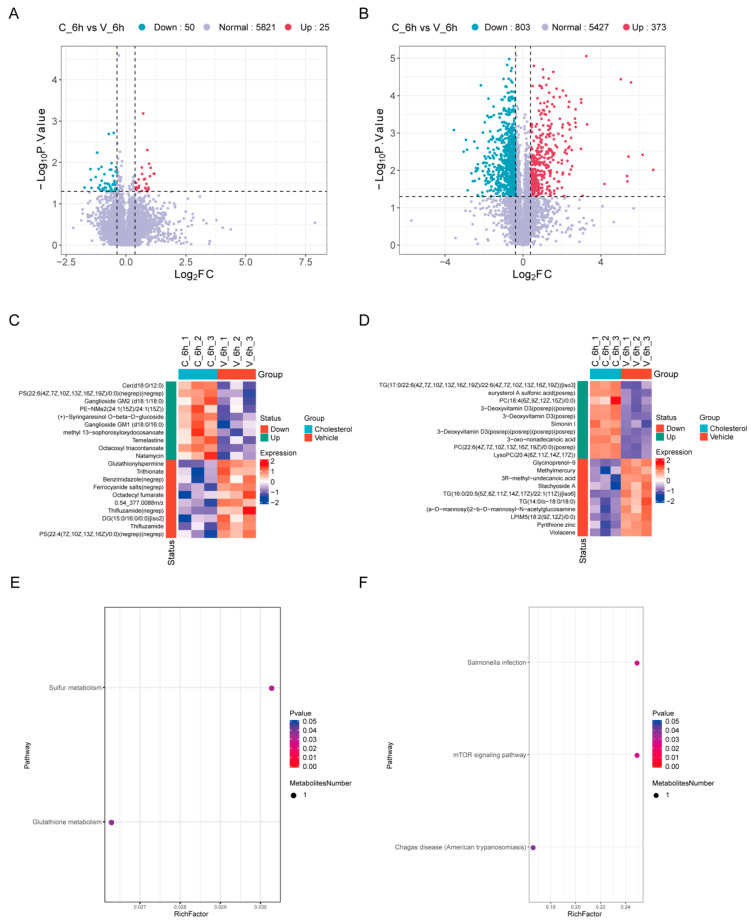
Lipidomic reprogramming in HeLa cells after cholesterol exposure. (**A**,**B**) Volcano plots of DAMs in the negative (**A**) and positive (**B**) ion modes. Each point represents a metabolite: blue = downregulated, red = upregulated, purple = no significant change. DAMs are defined by |log2FC| > 0.38 (1.3-fold) and *p*-value < 0.05. (**C**,**D**) Heatmaps of DAMs in the negative (**C**) and positive (**D**) ion modes. Rows = metabolites, columns = samples (cholesterol/vehicle groups). Color intensity reflects the expression ratio. (**E**,**F**) KEGG pathway enrichment of DAMs in the negative ion mode (**E**) and positive ion mode (**F**).

**Figure 5 ijms-26-07108-f005:**
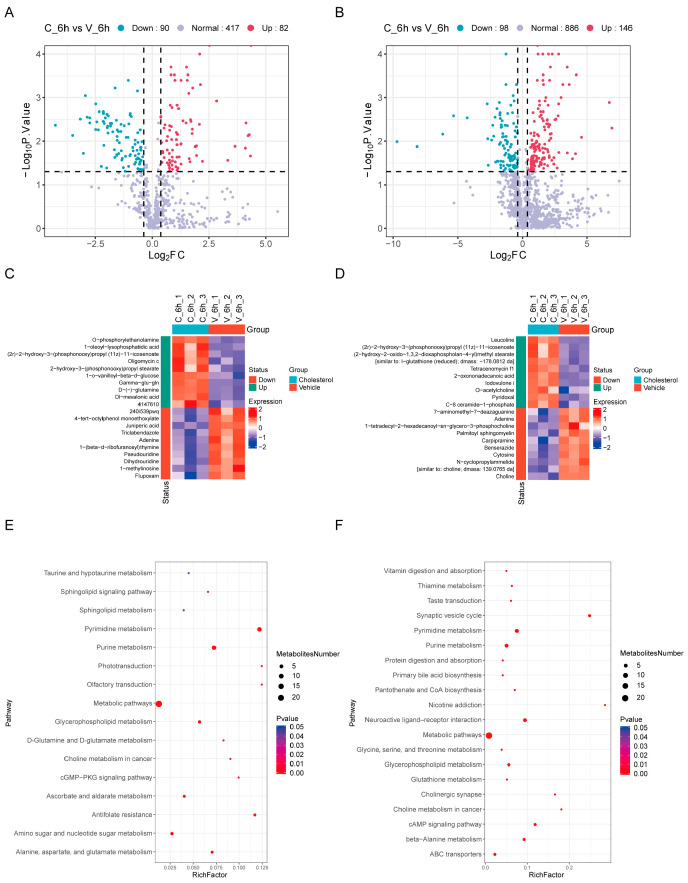
Broad spectrum metabolomic alterations in HeLa cells after cholesterol exposure. (**A**,**B**) Volcano plots of DAMs in the negative (**A**) and positive (**B**) ion modes. Each dot represents a metabolite: blue = downregulated, red = upregulated, and purple = no significant change. DAMs are defined by |log2FC| > 0.38 (1.3-fold) and *p*-value < 0.05. (**C**,**D**) Heatmaps of DAMs in the negative (**C**) and positive (**D**) ion modes. Rows = metabolites, columns = samples (cholesterol/vehicle groups). Color intensity reflects expression ratio. (**E**,**F**) KEGG pathway enrichment of DAMs in the negative ion mode (**E**) and positive ion mode (**F**).

**Figure 6 ijms-26-07108-f006:**
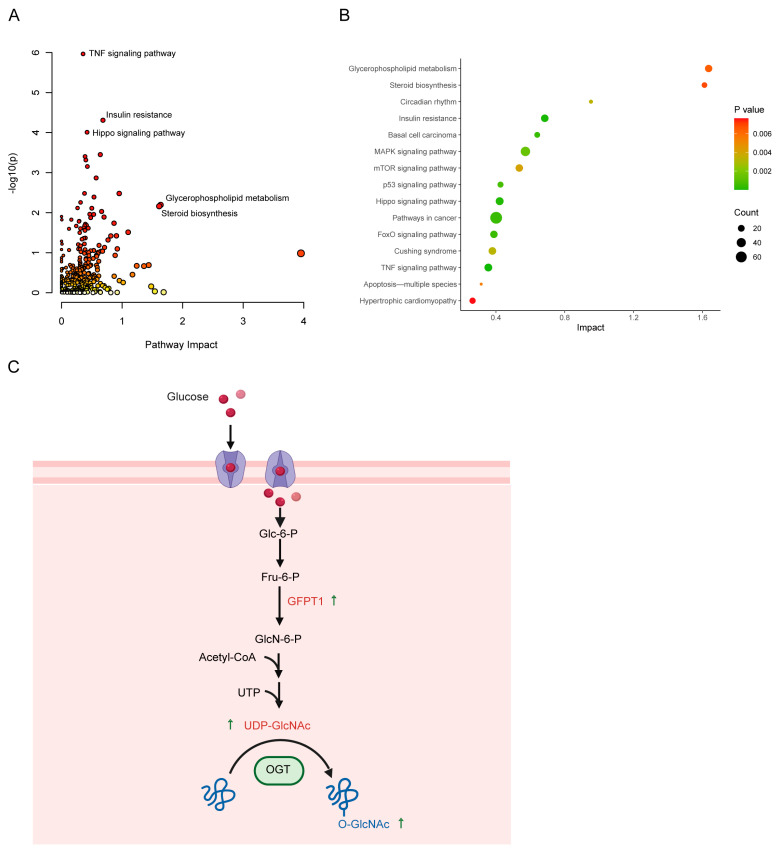
Integrated transcriptomic and metabolomic analysis of cholesterol-mediated metabolic reprogramming in HeLa cells. (**A**) Enrichment analysis of key pathways altered by cholesterol treatment. Larger circles indicate greater pathway enrichment, while darker colorsdenote higher statistical significance. (**B**) Bubble plot visualizing pathway enrichment results. (**C**) A schematic diagram of UDP-GlcNAc generation dependent on GFPT1 and the protein O-GlcNAc modification signaling pathway. Black straight arrows indicate the direction of metabolic conversion (straight downward), the upregulation of a gene or metabolite are shown by green arrows adjacent to labels). The curved arrow represents the O-linked N-acetylglucosamine transferae (OGT) catalyzed transfer of O-GlcNAc from UDP-GlcNAc to target proteins.

**Figure 7 ijms-26-07108-f007:**
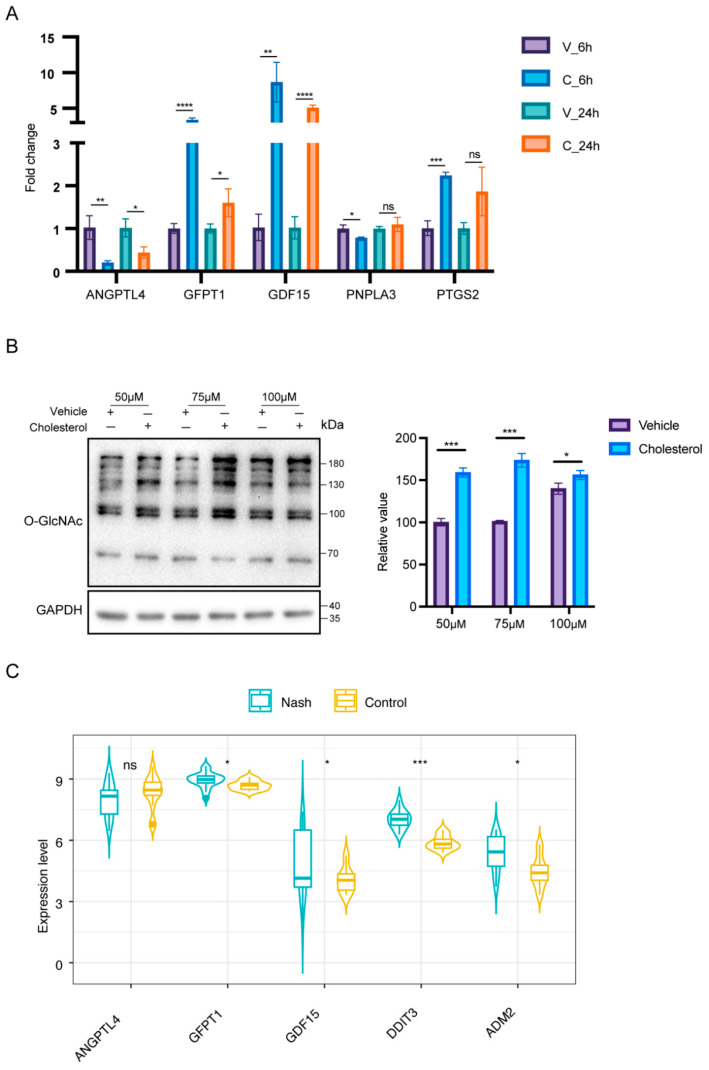
Gene expression validation and clinical correlation analysis. (**A**) mRNA expression of cholesterol processing in HeLa cells at 6 h and 24 h. (**B**) HeLa cells were cultured with different concentrations of CHOL (50 to 100 μM) or vehicle (the same concentration of MβCD used to dissolve cholesterol) in 10% FBS-containing medium for 6 h. Then cells were harvested for Western blotting. (**C**) mRNA levels in the GSE126848 cohorts of NASH. ns, not significant; * *p* < 0.05; ** *p* < 0.01; *** *p* < 0.001; **** *p* < 0.0001.

## Data Availability

Data is contained within the article and Appendix A.

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
