# Peer review of "Integrated Transcriptome and Metabolome Analyses Uncover Cholesterol-Responsive Gene Networks"

_ijms, 2025, doi:10.3390/ijms26157108_

Round 1
Reviewer 1 Report
Comments and Suggestions for Authors
The manuscript aimed to uncover cholesterol-responsive gene networks through integrated transcriptome and metabolome analyses. The manuscript is novel and interesting, but the authors should consider the following suggestions to improve its quality.
- The Abstract section should briefly introduce the experimental design scheme and the specific test results. In addition, the key words are inappropriate.
-
In the Introduction section, a brief overview of the research significance on the application of HeLa cells in in vitro experiments should be provided.
-
In the Results section, the content related to the experimental design should be placed in the Materials and Methods section.
Reviewer 2 Report
Comments and Suggestions for Authors
Cholesterol is fundamental to many aspects of cellular biology, and its dysregulation contributes to a large number of prevalent diseases, highlighting the need to understand cholesterol regulation better. Research to date focuses on isolated pathways and single-omics approaches, which do not fully capture the complexity and dynamics of the relevant processes in cholesterol metabolism. The authors present a clear and compelling need for their approach, a multi-omics strategy including both RNAseq and lipidomics.
The paper is well-written and mostly clear in its presentation of complex data. Conclusions are pointed out and explained so that the reader has some take-home points to ponder. There are times when the text seems bogged down in numbers and abbreviations whose significance is unclear. Below are some suggestions for an improved manuscript:
- 2.1.2 points out CPA2 as a potential mediator. However, its upregulation and confidence value seems very minimal, no greater than a great number of other genes in this dataset. It’s mention and labeling seems unwarranted.
- 2.1.2. The last sentence of this seems to prematurely state results that are actually discussed in the following section. Consider deleting.
- A number of abbreviations are undefined, at least from my view. GSEA, WGCNA, PPI. In many cases it would help the reader to understand significance if gene names were more than an abbreviation.
- Figure 2 might be reorganized so that it fits on one page or is 2 separate figures. Figure S2 also seems to need some minimal reformatting to place the panels of C in a horizontal arrangement.
- 2.1.3. “FOXO transcription factor binding motifs were significantly enriched…” How do you know this? No data is referred to to support this statement.
- 2.1.4. I found this section to be very confusing to read and understand. I believe that some greater explanation of terms and/or reorganization might improve its comprehension, as currently the term ‘module’ and the colors and numbers referencing these ‘modules’ is somewhat cryptic.
- 2.2.1. Page 14. Top paragraph includes a number of words that seem to be unnecessarily vague:
- “significantly altered”. Up or down?
- “group-specific trends”. What kind of trends?
- “drug interactions”. What drug was used in this study?
- “differentially expressed”. Up or down?
- 2.4. While the complex omics data will have much use to the scientific community, I was particularly pleased that there was some specific qPCR and western blotting validation of the omics data, which can be prone to error. Unfortunately, the authors skimmed over this data way too quickly. Please consider your experiments and rationale in more detail in this section.
- Methods. Identity and source of western blotting antibodies should be stated; sequence of qPCR primers should be included.
